# Structures of Echovirus 30 in complex with its receptors inform a rational prediction for enterovirus receptor usage

Kang Wang [1,2,3,8], Ling Zhu [1,8], Yao Sun [1,8], Minhao Li [4,5], Xin Zhao [6], Lunbiao Cui [2], Li Zhang [2], George F. Gao [6], Weiwei Zhai [4,7], Fengcai Zhu [2✉], Zihe Rao[1,3] & Xiangxi Wang [1,3✉]

Receptor usage that determines cell tropism and drives viral classification closely correlates with the virus structure. *Enterovirus B* (EV-B) consists of several subgroups according to receptor usage, among which echovirus 30 (E30), a leading causative agent for human aseptic meningitis, utilizes FcRn as an uncoating receptor. However, receptors for many EVs remain unknown. Here we analyzed the atomic structures of E30 mature virion, empty- and A-particles, which reveals serotype-specific epitopes and striking conformational differences between the subgroups within EV-Bs. Of these, the VP1 BC loop markedly distinguishes E30 from other EV-Bs, indicative of a role as a structural marker for EV-B. By obtaining cryo-electron microscopy structures of E30 in complex with its receptor FcRn and CD55 and comparing its homologs, we deciphered the underlying molecular basis for receptor recognition. Together with experimentally derived viral receptor identifications, we developed a structure-based in silico algorithm to inform a rational prediction for EV receptor usage.

[1] CAS Key Laboratory of Infection and Immunity, CAS Center for Excellence in Biomacromolecules, Institute of Biophysics, Chinese Academy of Sciences, Beijing 100101, China. [2] NHC Key Laboratories of Enteric Pathogenic Microbiology, Jiangsu Provincial Center for Disease Control and Prevention, Nanjing 210009, China. [3] State Key Laboratory of Medicinal Chemical Biology, College of Life Sciences and College of Pharmacy and Drug Discovery Center for Infectious Diseases, Nankai University, Tianjin 300353, China. [4] Key Laboratory of Zoological Systematics and Evolution, Institute of Zoology, Chinese Academy of Sciences, Beijing, China. [5] University of Chinese Academy of Sciences, Beijing 100049, China. [6] CAS Key Laboratory of Pathogenic Microbiology and Immunology, Institute of Microbiology, Chinese Academy of Sciences, Beijing 100101, China. [7] Center for Excellence in Animal Evolution and Genetics, Chinese Academy of Sciences, Kunming, China. [8] These authors contributed equally: Kang Wang, Ling Zhu, Yao Sun. ✉email: jszfc@vip.sina.com; xiangxi@ibp.ac.cn

The genus *Enterovirus* within the *Picornaviridae*, a family of non-enveloped viruses with a positive single-stranded RNA genome, is comprised of more than 100 serotypes[1]. Human enteroviruses (EV) are presently classified into four groups, *Enterovirus* A-D (EV-A, B, C, and D), among which EV-B, the largest group, consists of over 60 serotypes, including all 6 serotypes of coxsackievirus B (CVB1-6), coxsackievirus A9 (CVA9), over 30 serotypes of echoviruses and more than 20 newly identified enteroviruses[2]. EV-Bs are the main causative agents of aseptic meningitis[3,4] and many serious acute diseases, including viral encephalitis and even acute diarrhea with echovirus 30 (E30) being amongst the most commonly circulating serotype. In recent years, severe outbreaks of E30 infections have been documented in America, Europe, and Asia[5]. Neonates and infants are at greatest risk of developing severe echovirus-induced diseases, and infection within the first few weeks of life can be fatal[3,4]. Currently, there are no approved vaccines or antiviral therapies available for treating infections caused by echoviruses. Although capsid structures for many EVs have been studied extensively[6–19], large gaps in our knowledge concerning determinants of specificity between the serotypes/subgroups and characteristics for receptor usage in EV-Bs still exist. Therefore, an in-depth understanding of E30 structural features and receptor recognition mechanisms should be useful in providing guidance for the rational drug design against EV-Bs infections.

EV-Bs share the same overall structure found in other picornaviruses with a ~7.5 kb single-stranded positive-sense RNA genome that encodes four viral protein subunits VP1-4, 60 copies of which assemble into a pseudo $T = 3$ icosahedral capsid[20]. While VP1-3 interact with each other to construct the outer architecture of the viral capsid, their N-termini, together with VP4, line the interior. Natural empty particles (without RNA, termed "E-particles") are often also produced, in which VP0 is not further processed into VP2 and VP4. Two fundamental configurations in EVs, the compact form, generally ~160S mature virions (termed "F-particles") and the expanded state with altered antigenic properties, e.g. ~135S uncoating intermediates (termed "A-particles") are well documented[8,18,19]. Interestingly, there have been cases where the natural empty particles are expanded; sometimes, however, the empty particles resemble the mature virus in structure and antigenicity[21,22]. Evidently, empty particles with virion-like antigenic features are of considerable interest for vaccine development.

EV entry involves two key steps: (I) attachment, in which the virus binds to attachment receptors on the surface of the host cell; and (II) uncoating, in which the viral genome is released from viral particles into host cells[23]. These two steps are either accomplished by two separate types of receptors (attachment and uncoating receptors) or single bi-functional receptor(s)[24–27]. A number of receptors have been identified and known to facilitate the entry of EVs, including scavenger receptor B2 (SCARB2; a receptor for EV71, CVA16 and a subgroup of EV-A), Kringle-containing trans-membrane protein (KREMEN1 for another subgroup of EV-A, including CVA10), coxsackie and adenovirus receptor (CAR; the uncoating receptor for one subgroup of EV-B, including all six serotypes of CVBs), CD55 (the attachment receptor for many EV-Bs), the newly identified Neonatal Fc Receptor (FcRn; the uncoating receptor for another major subgroup of EV-B, containing E30), Intercellular adhesion molecule 1 (ICAM1; the uncoating receptor for one subgroup of EV-C, including CVA21 and CVA24v) and CD155 known as PVR (poliovirus receptor for EV-C)[17,28–32]. Recently, a number of atomic structures of enteroviruses in complex with their receptors, including EV71, CVA10, E6, PV1 and CVA24v have been characterized[28,33–36]. Most of the uncoating receptors bind to the "canyon" sites of the viral particle, dislodging the "pocket factor"

in VP1 and subsequently triggering conformational changes of the intact particle to release the genome[28,33,34,36–40], albeit that SCARB2 binds EV71 outside the canyon. Receptor usage correlates with the capsid structure, indicative of receptor switching driving viral evolution and subsequent viral classification. Consequently, viral receptor usage is a determinant for cell tropism and its evolution can change virus targets at the level of cells to host ranges. Although virus–receptor interactions have been well characterized for several EVs, there remain a large number of viruses, especially those in more recently identified genera, whose receptors are yet to be identified.

Here we report the structural characterizations of E30 E-particle, A-particle and mature virion, among which E-particle and mature virion are compact with fully ordered epitopes, showing the potential to be ideal vaccine candidates. Structural comparisons of E30 with EV-As, other EV-Bs and EV-Cs reveal serotype-specific epitopes and specific differences between the subgroups. We also determined the atomic structures of E30 in complex with its receptor FcRn and CD55. Systematical comparisons of E30-receptors with other representative EVs and their receptors pinpoint key structural elements modulating the molecular basis for receptor recognition. This information in combination with experimentally derived viral receptor identifications, form the basis for the development of a structure-based in silico algorithm for the prediction of EVs' receptor usage.

## Results

**Characterization and structure determination.** Isolated E30 was plaque-purified and propagated in human rhabdomyosarcoma (RD) cells for purification as previously described[18]. We separated out two distinct types of particles by ultracentrifugation and characterized them using SDS-PAGE, single-stranded RNA detection and analytical ultracentrifugation as either 143S F-particles, containing RNA and a full complement of proteins VP1–VP4, or 80S E-particles containing VP0, VP1, and VP3 (Supplementary Fig. 1). The sedimentation coefficient of the F-particles was considerably smaller than those of most EV's mature virions (~150S), suggesting the F-particles might be mixed with a low ratio of A-particles, as previously observed in CVA10[8]. The purified E30 particles were recorded and visualized using Cryo-EM. In line with the sedimentation analysis, three types of particles: F-particle (~50%), A-particle (~7%) and E-particle (~43%) were separated when applied with no-alignment 3D classification and were reconstructed to 2.9 Å, 2.9 Å and 3.4 Å, respectively (Fig. 1 and Supplementary Fig. 2) based on the "gold" standard Fourier shell correlation (FSC) = 0.143 criterion[41] using Relion 3.0[42] (Supplementary Fig. 3). The density maps of all three types of particles show the clear delineation of protein backbones and resolve side chains of most residues, which allowed us to build atomic models and these models were refined and validated using standard X-ray crystallographic metrics (Fig. 1a–d and Supplementary Fig. 4 and Supplementary Table 1).

**Overall structures of E30 F-particle, A-particle, and E-particle.** The three types of particles reveal a pseudo $T = 3$ symmetry arrangement and the overall structures of three particles resemble those of other EVs (Fig. 2a). The capsid proteins (VP1, VP2, and VP3) adopt the classical eight-stranded anti-parallel β-barrel configuration with the N-termini residing inside and the C-termini exposed on the surface. Apart from some disorder on the inside of the E-particle, the structures of full and empty particles are mostly very similar, with the external surface expected to be antigenically indistinguishable (Fig. 1a). The investigation of E30 immunogenicity by vaccinating mice separately with purified F- and E-particles show high neutralization tilters, indicating that

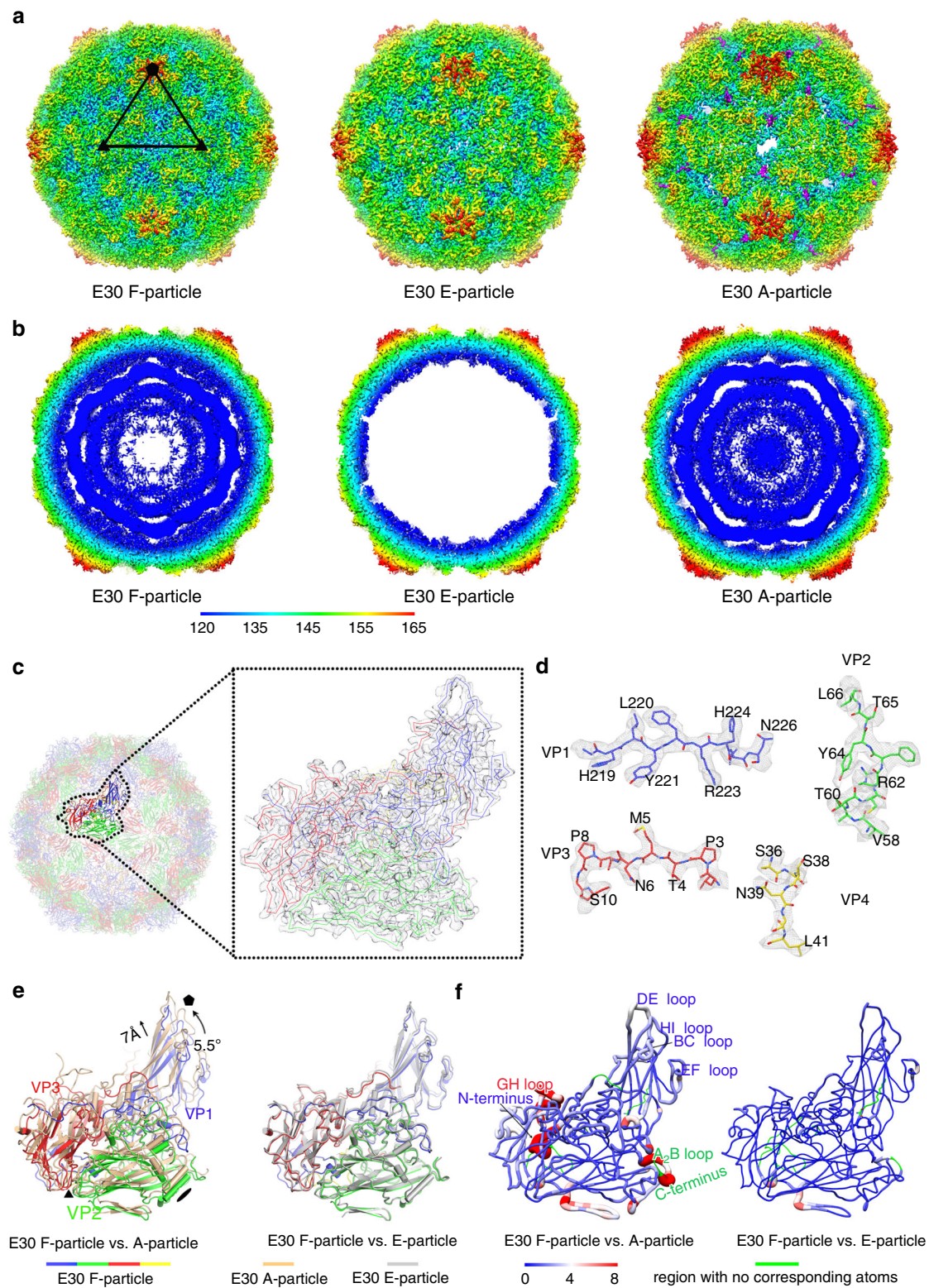

both F- and E-particles are highly immunogenic and can elicit high neutralizing antibody (NAb) titers, which is also consistent with the structural analysis (Supplementary Fig. 5) [see coordinated submission by Wang et al.[43]]. Like most enteroviruses, E30 F-particle possesses the hydrophobic pocket in VP1 harboring a pocket factor, whereas both E- and A-particles lack hydrophobic pocket and harbor no pocket factor. Compared to F- and E-particles, the E30 A-particle is markedly expanded with a ~4.5%

increase in radius with a maximum diameter of ~345 Å along the fivefold axis (Fig. 1a, b). The expansion of the A-particle reflects tectonic movements in the particle, forming perforations at the icosahedral two-fold axes, accompanied by a ~5.5° counterclockwise rotation of the protomeric unit near the threefold axis and a ~7 Å shift away from the two-fold axis of the protomer unit (Fig. 1e). Superposition of the asymmetric units (protomers) verifies the similarity in capsid structures of the F- and E-particles

**Fig. 1 Cryo-EM structures of E30 F-, E-, and A-particle. a** Surface representation of E30 F-, E-, and A-particle along the icosahedral twofold axes. The surfaces are rainbow-colored by radius from blue through yellow, green, and indigo to red, which correspond to 120, 135, 145, 155, and 165 nm, respectively. One icosahedral asymmetric unit is marked with a black triangle in E30 F-particle and icosahedral symmetry axes are drawn in black. The obstructed off-axis channels near the twofold axes in A particle are highlighted in magenta. **b** Thin slices of the central sections of each particle in a; they were colored with the same color scheme. **c** Cartoon representation of the E30 F-particle perceived from the twofold axes (left). A single icosahedral protomer is zoomed in, shown as line and accentuated in inset (right). The signature colors (VP1, blue; VP2, green; VP3, red; VP4, yellow) were applied to each subunit. **d** Electron density maps from a section of VP1, VP2, VP3, and VP4, respectively. **e** Protomeric units shown with respect to the icosahedral axes of the particles by superposing whole particles. The F-, A-, and E-particle were colored in signature color, wheat and gray, respectively as the color bars below display. **f** Protomeric superpositions of F- and A-particle (left), as well as F- and E- particle (right). Structural differences are mapped onto the protomer of the full particle; the thickness and color of the worm representation reflect the local deviation between the structures [from blue (0 Å) to red (8 Å)]. Regions missing in the A- or E-particle are shown in green (VP4 is omitted).

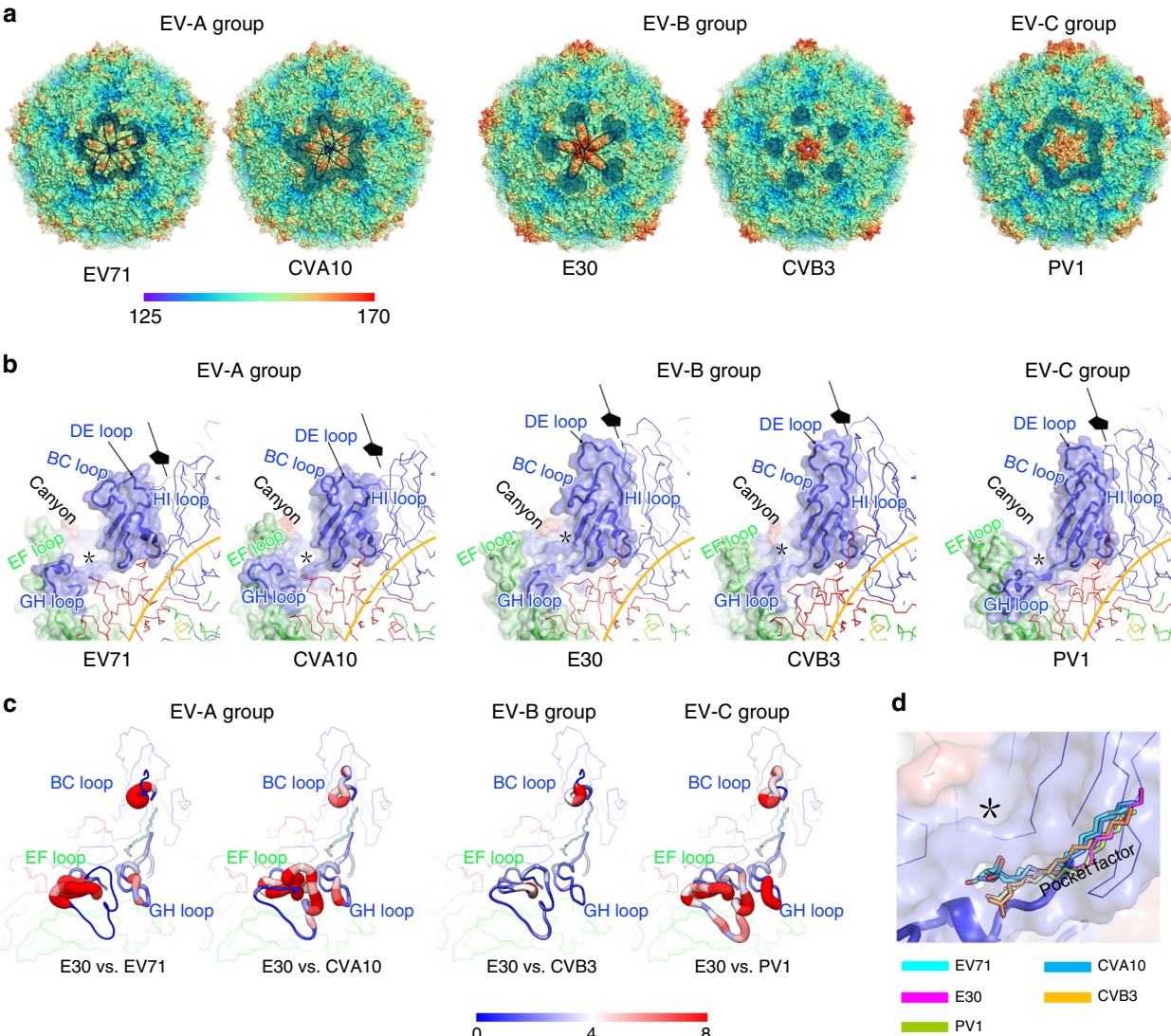

**Fig. 2 Structure comparisons. a** Comparison of capsid from E30 F-particle with those from representatives of enterovirus A (EV71-3VBS, CVA10-6AKS), B (E6-6ILP), and C (PV1- 1HXS) species. Capsids are rainbow-colored by radius from blue to red as shown in the color bar below. The canyon areas around fivefold axes are highlighted by shadow and the 'star-shaped' protrusions surrounding the mesa are outlined with black dotted lines. **b** Surface of the biological protomer (VP1, blue; VP2, green; VP3, red) of representative particles. Loops forming the canyon walls (VP1 BC loop, VP1 GH loop, and VP2 EF loop) and "mesa" structures (VP1 DE loop) are labeled in corresponding colors. Hydrophobic pocket from each particle is marked with a black star. **c** Comparison of canyon from E30 F-particle with those from representative particles. As the same with 1f, structural differences are mapped onto the protomer of E30 F-particle; the thickness and color of the worm representation reflect the local deviation between the structures (from blue (0 Å) to red (8 Å)). **d** Comparison of hydrophobic pocket of E30 with those from representative particles. Pocket factors from representative viruses were aligned together within the hydrophobic pocket of E30. The color scheme adopted in (**c**, **d**) is the same as that in (**b**).

(RMSD = 0.4 Å in Cα atoms). However, a close examination of the particles reveals that the F-particle differs significantly from the A-particle (RMSD = 1.1 Å in Cα atoms) at the VP1 N-terminus, VP3 GH loop and the A$_2$B loop and C-terminus of VP2 (Fig. 1f and Supplementary Fig. 6). These conformational changes of individual capsid proteins relay a cascade of alterations, separating the α helices of adjacent VP2 subunits and leading to large perforations at the two-fold axes in the A-particle, through which the viral RNA and VP4 might exit (Supplementary Fig. 6).

**Structural basis for distinguishing the groups within the EVs.** Compared to EV-As and EV-Cs, EV-Bs exhibit more prominent and condensed plateaus (called "mesa") constructed by the longer and raised DE loop in VP1 around the fivefold axes (Fig. 2a and Supplementary Fig. 7), similar to the "cooling tower" structures observed in human Aichi virus, a picornavirus in the genus *Kobuvirus*[44]. Interestingly, E30 possesses five additional continuous and "star-shaped" protrusions surrounding the mesa, characterized as specific for EV-As[8], although the overall structure of E30 most closely resembles those of other EV-Bs (Fig. 2a). These protrusions arise from the loops near the five-fold axes, in particular the VP1 BC loop, which assumes a slightly longer and raised conformation when compared to their counterparts from other EV-Bs (Fig. 2b and Supplementary Fig. 8). Like other EVs, E30 contains the canyon-like depression and the hydrophobic pocket in VP1 harboring a pocket factor beneath the depression (Fig. 2b–d). However, compared with EV-As and EV-Cs, which have a continuous circular canyon around each five-fold mesa, in EV-Bs, there are, instead, five distinct ~15-Å deep depressions. A ridge between each depression is formed by the C-termini of VP1 and VP3 (Fig. 2a). A number of key structural elements contributing to the canyon, including the VP1 BC loop (the north wall of the canyon), the VP2 EF loop (back part of the south wall) and significantly shortened VP1 GH loop (front part of the south wall) adopt different conformations and move toward the front, together with the ridges, ablating the back portion of the canyon (Fig. 2b, c). In line with the diverse morphologies of the canyon determined by various configurations of VP1 BC, VP1 GH, and VP2 EF loops, different subgroups of EVs utilize different uncoating receptors for initiating efficient entry[17,28–30,36]. In addition, aforementioned three loops, the VP1 EF loops, and VP1 C-terminus have been widely reported to be dominant epitopes, which antibodies target possibly using a receptor mimic mechanism to neutralize virus infection[7,45,46]. Perhaps correlated with the selective pressures of receptor usage and escape from host immune responses, the VP1 BC, VP1 GH, and VP2 EF loops are among the most diverse regions in both sequence and conformation (Fig. 2c and Supplementary Figs. 7 and 8). However, in comparison with EV-As and EV-Cs, the VP1 GH, and VP2 EF loops seem relatively conserved, while the VP1 BC loop is more hypervariable in EV-Bs (Fig. 2c and Supplementary Figs. 8 and 9). Thus, the VP1 BC loop might act as a structural marker to distinguish serotypes in EV-Bs while VP1 GH and VP2 EF loops can be rationally targeted to obtain broad neutralizing antibodies against most EV-Bs [see coordinated submission by Wang et al.[43]]. Additionally, the pocket factor in EV-As is partly exposed on the floor of the canyon, distinguishing EV-As from other EVs (Fig. 2d).

**Structures of E30 in complex with its receptors.** Recent studies reported CD55 and FcRn function as attachment and an uncoating receptors, respectively for many EV-Bs[17,26,28,47]. We determined the cryo-EM structures of E30 F-particle in complex with FcRn and CD55 under neutral conditions at 3.3 Å and 3.6 Å, respectively and obtained atomic models for these two complexes

(Fig. 3a and Supplementary Figs. 2 and 3). Bindings of FcRn and CD55 have minimal impact on the capsid structure with RMSDs of 0.3 Å and 0.4 Å, respectively between mature and complexed viruses. However, in FcRn-bound viruses, the pocket factor was released and the pocket partially collapsed with the side chains of Tyr147, Asn215, and Met217 repositioning to decrease the pocket volume, similar to the pocket observed in E-particle, whilst there is no notable reduction in the amount of the pocket factor in CD55-bound viruses (Supplementary Fig. 10), indicative of that FcRn is capable of triggering viral pocket factor dislodgement, but CD55 might only facilitate viral attachment at neutral pH. It's worthy to note viral RNA is still inside the capsid in FcRn-bound virus (Supplementary Fig. 3e), suggesting extra factors/signals are required to trigger RNA release. Despite having a close phylogenetic relationship with E30, E6 exhibits clearly a low pH-dependent dislodge of the "pocket factor" mediated by FcRn[28], revealing different patterns of uncoating processes. FcRn inserts into the canyon depression of E30 through the α2 and α3 helices. The FcRn-binding site on E30 comprises residues from the VP1 BC, VP1 EF, VP1 GH, and VP2 EF loops, which form the north and south rims of the canyon and bear antigenic residues (Fig. 3b, c) [see coordinated submission by Wang et al.[43]]. The VP1 BC loop is longer in E30 than in most EV-Bs (for example, 4 residues longer than in CVB3), and residues Asp86 and Glu87 in this loop, together with residue Asp138 from VP2 EF loop establish charge interactions with residues Lys80, Arg140, and Lys150 from FcRn (Fig. 3d). Overall, hydrophilic interactions dominate the tight associations (with an interaction area of ~1150 Å$^2$) between E30 and FcRn, where 14 viral and 14 FcRn residues are involved in forming 16 hydrogen bonds (Fig. 3c and Supplementary Table 2). Compared to FcRn, the interactions between E30 and CD55 are reduced by nearly half, where the short consensus repeat (SCR) 3 of CD55 contacts both VP1 GH and VP2 EF loops in E30 via hydrophilic interactions (Fig. 3c and Supplementary Table 3). In addition, CD55 presents a distinct attachment pattern to E30 in that the binding site lies outside the canyon, extending to the "south" (Fig. 3e). This site is roughly similar to those observed for SCARB2 binding to EV71 and for integrin binding to foot and mouth disease virus[33,38]. Nevertheless, most uncoating receptors, except for SCARB2, adopt an in-canyon attachment strategy to dislodge the pocket factors in enteroviruses, whereas attachment receptor binding is more diverse.

**Structural elements modulating uncoating receptor recognition.** Correct engagement with a functional receptor is critical to viral infectivity, and is also the determinant of viral species specificity. Currently, at least seven uncoating receptors [SCARB2, KREMEN1, FcRn, CAR, CD155, intercellular adhesion molecules 1 and 5 (ICAM-1 and ICAM-5, uncoating receptors for CVA24 and EV-D68, respectively)] have been identified for human EVs, together with the availabilities of structures of human EVs complexed with six receptors, excluding the unavailable complex structure of EV-D68 and ICAM-5[17,28–30,36,48,49], allowing us to map key structural elements modulating receptor recognition. Except for SCARB2 which binds EV71 outside the canyon, other five receptors insert into the canyon at different areas. Given that a number of key loops, such as the VP1 BC, EF, GH loops, and VP2 EF loop, contribute to the formation of the canyon and determine specific differences between the groups/subgroups, some of these loops modulate recognition of various receptors, controlling the receptor usage (Fig. 4a). Surprisingly, VP1 GH and VP2 EF loops of EV71 still dominate SCARB2-virus interactions, as is observed for the in-canyon attachment of most other enteroviruses, suggesting a conserved mechanism by which receptor binding in or near the canyon might facilitate viral

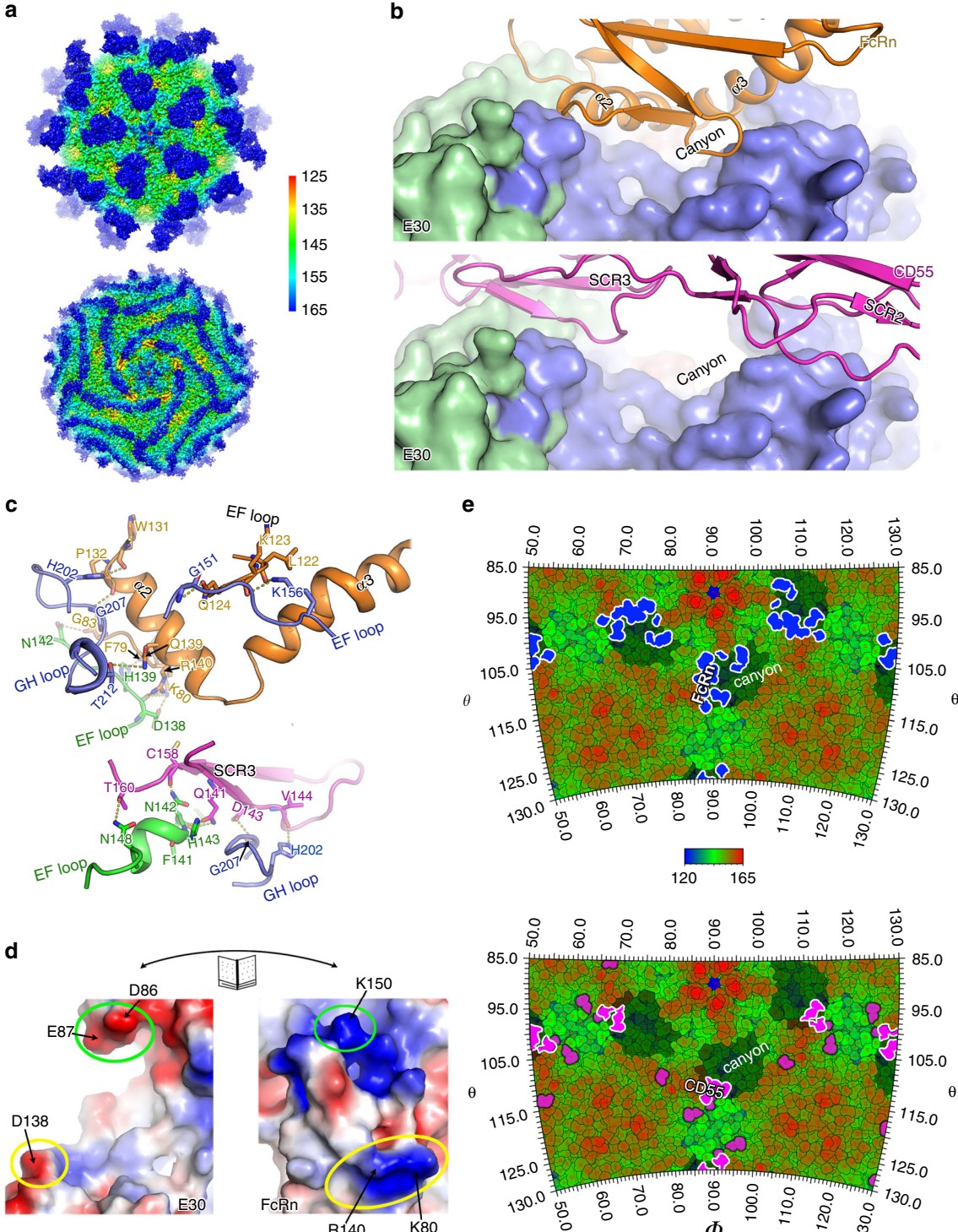

**Fig. 3 Structural details of the interaction between E30 and its uncoating (FcRn) or attachment (CD55) receptor. a** Surface representation of E30-FcRn complex (up) and E30-CD55-complex (down) along the icosahedral fivefold axes. The surfaces are colored by radius as shown in the color bar right. **b** Platform for receptors (FcRn, top and CD5, bottom) binding with E30. The FcRn colored in orange and CD55 colored in magenta are shown in cartoon representation, and viral protomers as surface representation are applied in the same color scheme as in Fig. 1c. **c** Residues at the main E30-FcRn (up) and E30-CD55 (down) interfaces. The side or main chains involved in mutual interactions are shown as sticks. The hydrogen bonds are shown as yellow dashed lines and the color scheme adopted here is the same as above. **d** Electrostatic surface representation of E30 and FcRn. The surfaces are colored in a gradient of blue to red in accordance with positive to negative charge, respectively. And the residues in E30 are labeled and circled in the same color in respect to their interacting partners in FcRn. **e** Roadmap exhibiting the footprints of FcRn (up) or CD55 (down) on the surface of E30. The roadmaps are radius-colored from blue (120 nm) to red (165 nm) with θ and φ representing latitude and longitude, respectively. The canyons surrounding the fivefold axes are shaded by shadow. The footprints of FcRn and CD55 are colored in blue and magenta, respectively, and the overlaps between the canyon and FcRn are circled in red lines and accentuated in dark blue.

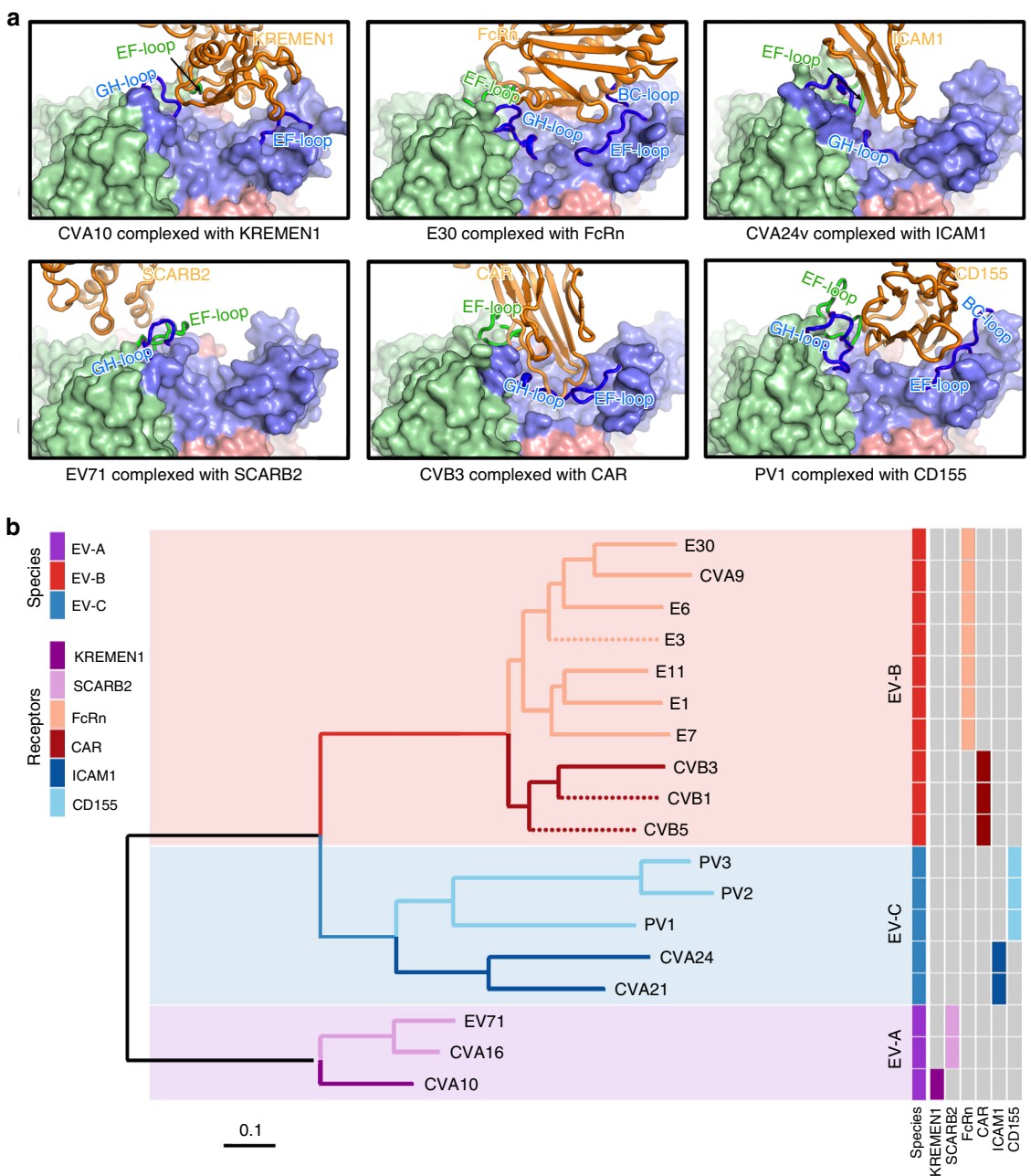

**Fig. 4 Structure-based receptor prediction of enteroviruses. a** Engagement mode between a protomeric unit of the representative members of species A (left), B (middle), and C (right) and its corresponding uncoating receptor. All the complexes (EV71-SCARB2, 6I2K; CB3-CAR, 1JEW; CVA24v-ICAM1, 6EIT; PV1-CD155, 3J8F) are shown from the identical perspective. The receptors are shown as cartoon in orange, while the protomers are shown as surface with loops directly interacting with their receptors shown as cartoon, accentuated in a darker color and traced in magenta (VP1 in blue, VP2 in green and VP3 in red). **b** Phylogenetic tree generated based on the structure-based in silico analysis of three key motifs: VP1 GH, VP2 EF, and VP1 EF loops. The dotted branches represent the new structures (pdb accession codes: 7C9X, 7C9Y, 7C9Z) that were selected for predication verification.

uncoating of the EVs in the endosome. Unexpectedly, a large number of residues within the binding footprint, also constituting key structural elements to mediate receptor recognition, are not conserved across receptor-dependent subgroups of human EVs (Supplementary Fig. 11). Therefore, we have investigated how detailed structural information on these key elements could be rationally analyzed to develop an in silico algorithm for predicting receptor usage in human EVs. Given that three key structural elements of the VP1 GH, VP2 EF, and VP1 EF loops

dominate virus–receptor interactions, we systematically calculated pairwise Euclidean distances between all amino acids for all possible $\binom{n}{2}$ pairs of these three key elements from 15 structure-available human EVs, which defines a k ($k = \binom{n}{2}$) dimensional space delineating the structural characteristics of the receptor-recognition sites. After that we obtained a $15 \times 15$ dimensional distance matrix and constructed the phylogenetic

tree of these 15 human EVs by using Neighbor joining algorithm from PAUP (see methods)[50]. As shown in Fig. 4b, the phylogenetic tree could clearly separate these human EVs into three groups: EV-A, EV-B, and EV-C. Within each group, viruses with distinct receptors were further grouped into mono-phylogenetic clades (Fig. 4b). This structure-based in silico analysis completely matches the results of experimentally derived receptor identification, therefore, it can be used to deduce the receptor usage for human EVs. To test the robustness of our structure-based algorithm, we determined the structures of three other human EVs (CVB1, CVB5 and Echovirus 3) and put these into our algorithm. The in silico analysis suggests that CVB1 and CVB5 are CAR-dependent human EVs and E3 utilizes FcRn as a functional receptor (Fig. 4b), predictions consistent with experimental reports[28,47,51]. These analyses suggest two lessons: (1) various binding sites (in or surrounding the canyon) revealed by different types of receptors on human EVs present distinct structural features and drive viral classification; (2) in turn, structural features of the human EV canyon and its surroundings provide inferences for receptor usages.

## Discussion

Receptor usage or the interplay between different receptors are likely to be important determinants for viral tropism and pathogenesis. Many enteroviruses employ one or more attachment receptors to efficiently achieve cell binding. However, most enteroviruses rely on an uncoating receptor to trigger the destabilizing rearrangements of capsid proteins, leading to the formation of a capsid mediated channel that connects the capsid two-fold pore and the endosomal membrane, through which the viral genome is released into the cytoplasm[8]. Although virus–receptor interactions have been characterized for several human EVs, there remains a large number of EVs, especially those in more recently identified EVs, for which receptors have not yet been identified. Generally, the standard verification of a bona fide viral receptor involves a number of assays, such as that expression of this receptor enables normally unsusceptible cell lines to support viral propagation; loss of expression of this receptor or its binding partner (e.g., antibodies) renders cells resistant to viral infection; soluble receptor could directly bind viral particles and neutralizes infection, but these assays are time-consuming. By contrast, the recent 'resolution revolution' in cryo-EM has accelerated the process for determining high-resolution structures of a wide array of previously intractable biological systems[52–58]. In this study, we provided models for E30 in complex with its receptors CD55 and FcRn and systematically analyzed available human EVs-receptor interactions, which pinpointed key structure-receptor usage correlates. We then developed a structure-based in silico algorithm that, in three test cases, successfully predicts the receptor usage previously reported from in vitro assays. This method provides an alternative option to unveil receptor usage for human EVs quickly.

In addition to causing viral encephalitis, viral meningitis, and viral meningo-encephalitis, EV-Bs are also the causative agents of acute flaccid paralysis, pneumonitis and hand-foot-and-mouth diseases, which are primarily caused by EV-As[59,60]. Furthermore, frequent natural recombination events between human EV species are a huge hindrance to the accurate prediction of the roles human EV subtypes might play during future outbreaks. Even though the polio and EV71 vaccines on the market efficiently prevent virus-specific caused diseases, so far, there is no approved drug or vaccine against other human EV infections. The fast in silico analysis of receptor usage for human EVs informs common characteristics for receptor-dependent subgroups. This information together with specific differences revealed here between the subgroups, provides guidance for the rational design of novel and effective multivalent and broad-spectrum vaccines against infections caused by human EVs.

## Methods

**Particle production and purification.** The E30 strain designated as JS2002 and isolated from a patient in Jiangsu Province of China was inoculated onto the RD cell monolayers seeded in 6-well plates at a multiplicity of infection (MOI) of 0.01. After incubation at 37 °C for an hour, the plates were rinsed with PBS (pH 7.4) three times and then covered with the agarose (2 mL/well) supplemented with 2% FBS. 72 h later, the plaques visible to naked eyes were picked up for cultivation. After another two rounds of such plaque-purifications, the acquired virus clone was inoculated into RD cells grown to 95% confluence at a MOI of 0.1. When 95% of the cells developed visible CPE 18–24 h post infection, the cultures were harvested and treated with freeze-thaw cycles three times. Then the solution was centrifuged at 1500 × g at 4 °C for 30 min to remove cell debris, and the supernatant was ultra-centrifuged at 120,000 × g for 2 h. The resulting pellets were resuspended in PBS and loaded onto a continuous 15–45% sucrose density gradient and centrifuged at 104,100 × g for 3.5 h. Three sets of fractions were collected and dialyzed against PBS buffer and the capsids from these fractions were imaged by negative staining transmission electron microscope (TEM) and cryo-EM.

**Negative stain.** Samples to be examined were diluted into PBS (pH 7.4) to an appropriate concentration (~0.5 mg/mL) and dropped onto a freshly glow-discharged carbon-coated grid. After rinsing twice with PBS, the grid was stained with 1% phosphotungstic acid (pH 7.0) and then loaded onto a FEI Tecnai Spirit 120-kV TEM for examination.

**Expression and purification of FcRn and CD55.** The soluble ectodomains of FcRn and CD55 were prepared as described previously[28]. Briefly, the constructs containing the coding fragments of human Fc fragment of IgG receptor and transporter (FCGRT) (A16-L282), and B2M were co-transfected into HEK293T cells for the expression of FcRn; and construct containing the coding fragment of human CD55 (D35-G285) was transfected into HEK293T cells to express CD55. All the sequences used here were synthesized by Genewiz and inserted into pCMV3 (Sino biological, China). Afterward, the supernatants were collected and purified by Ni-nitrilotriacetic acid (NTA) chromatography and then Hiload 16/600 Superdex 200 PG column (GE) gel chromatography with the buffer containing 20 mM Tris-HCl and 150 mM NaCl (pH 8.0).

**Cryo-EM and data collection.** For cryo-grids preparation, on the one hand, the purified E30 F-particles were mixed with the E30 E-particles to obtain the E30 mixtures, on the other hand, the purified E30 F-particles were mixed with FcRn or CD55 (at a ratio of ~1:200) on ice for 10Seconds to obtain the E30 F-particle-receptor mixtures. Afterward, 3 µl aliquot of the E30 mixtures or the mixture of E30 F-particle with FcRn, or the mixture of E30 F-particle with CD55 were deposited onto the freshly glow-discharged 400-mesh holey carbon-coated gold grid (Quantifoil, R 1.2/1.3, Jena). Grids were blotted for 3S in 100% relative humidity for plunge-freezing (Vitrobot; FEI) in liquid ethane. Cryo-EM data sets of E30 particles and E30-receptor complexes were collected with a Titan Krios microscope (FEI). In each case, movies (25 frames, each 0.2S, total dose of 30 e⁻ Å⁻²) were recorded using a Gatan K2 Summit detector with a defocus range of 1.2–2.5 µm. Automated single-particle data acquisition was performed by SerialEM, with a calibrated magnification of 59,000 yielding a final pixel size of 1.35 Å.

**Image processing.** A total of 1404, 1084, and 537 micrographs were recorded for E30, E30-FcRn-complex and E30-CD55-complex, respectively. Out of these, 957, 892, and 452 micrographs with visible CTF rings beyond 1/5 Å in their spectra were selected for further processing. The defocus value for each micrograph was determined using Gctf[61]. Then particles were picked and extracted for two-dimensional alignment, ab initio 3D model generation, three-dimensional auto-refinement, three-dimensional no-alignment classification, generating the final models of full particle, empty particle, A particle, E30-FcRn-complex and E30-CD55-complex by using 19,272, 16,676, 2406, 7299, and 1016 particles, respectively. After the high-resolution refinement and postprocessing (estimate the B-factor automatically), the final resolution was evaluated on the basis of the gold-standard Fourier shell correlation (threshold = 0.143)[41]. All procedures above were performed using Relion 3.0[42]. The local resolution was evaluated by ResMap[62]. Details of the data sets and refinement statistics are summarized in Supplementary Table 1.

**Model building and refinement.** The crystal structure of CVA9 (Protein Data Bank (PDB) ID: 1D4M) was manually fitted into the refined map of E30 full particle using Chimera[63] and corrected according to the amino sequence of E30 using COOT[64]. The atomic model was built after iterative positional B-factor refinement in real space using PHENIX[65] and re-adjustment in COOT. Structures of E30-empty particle, A particle as well as the viral portions of the E30-FcRn-complex and

E30-CD55-complex were determined using the same strategy. Likewise, the structures of FcRn and CD55 were corrected from the corresponding portions of previously determined E6-FcRn-complex (PDB ID: 6ILM) and E6-CD55-complex (PDB ID: 6ILK) and further combined with the viral portion structure to finally obtain the atomic structures of the complexes.

**Structure-based in silico analysis**. For each viral protein, we first obtained physical locations of all the amino acids (alpha-carbon) in the three-dimensional space. Then, we focused on three key motifs: VP1 EF, GH, and VP2 EF loops for the subsequent analysis. For a given protein region (VP1 EF, GH and VP2 EF loops), pairwise Euclidean distances between all amino acids were first calculated for all possible $\binom{n}{2}$ pairs within that region. These pairwise distances can define a k ($k = \binom{n}{2}$) dimensional space delineating the structural characteristics of the target region. For simplicity, we constrained the calculation to those homologous amino acid positions in the sequence alignment across 18 human EVs (i.e., excluding gap regions). In order to measure the distance between all these 18 viruses, we computed the distance between pairs of viruses as the root-mean-square deviation (RMSD) and between the k dimensional vectors. By computing all pairwise distances, we obtained a $18 \times 18$ dimensional distance matrix and constructed the phylogenetic tree of these 18 human EVs by using Neighbor joining algorithm from PAUP[50].

**Analytical ultracentrifugation (AUC)**. Sedimentation velocity experiments were performed on a Beckman XL-I analytical ultracentrifuge at 20 °C. Samples containing full and empty particles were diluted with PBS buffer (pH 7.4) to 400 μL with A280 nm absorption of about 0.7 and further loaded into a conventional double-sector quartz cell and mounted in a Beckman four-holeAn-60Ti rotor. Data were collected at $8064 \times g$ at a wavelength of 280 nm and the interference sedimentation coefficients were calculated using the SEDFIT software program (www.analyticalultracentrifugation.com).

**Particle stability thermal shift assay**. PaSTRy was performed with SYTO9 and SYPROred (both Invitrogen, Carlsbad, USA) as fluorescent probes to detect the exposed RNA and hydrophobic residues of the proteins by using an MX3005 qPCR instrument (Agilent, Santa Clara, USA). Here, we set up a 50 μl-reaction system which contained 2 μg of virus samples, 5 μM of SYTO9 and 3X SYPROred, and ramped up the temperature from 25 °C to 99 °C. Fluorescence was recorded in triplicate at an interval of 1 °C.

**Reporting summary**. Further information on research design is available in the Nature Research Life Sciences Reporting Summary linked to this article.

## Data availability

The atomic coordinates of E30 F-, E-, A-particles, E30-FcRn-complex, E30-CD55-complex, E3, CVB5, and CVB1 have been submitted to the Protein Data Bank with accession numbers: 7C9S, 7C9U, 7C9T, 7C9V, 7C9W, 7C9X, 7C9Y, and 7C9Z respectively. The cryo-EM density maps of E30 F-, E-, A-particles, E30-FcRn-complex, E30-CD55-complex, E3, CVB5, and CVB1 have been deposited in the Electron Microscopy Data Bank under accession codes: EMD- 30315, EMD- 30317, EMD- 30316, EMD- 30318, EMD- 30319, EMD-30320, EMD-30321, and EMD-30322, respectively. Other data are available from the corresponding authors upon reasonable request. Source data are provided with this paper.

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

## Acknowledgements

We thank B. Zhu, X. Huang and G. Ji for Cryo-EM data collection at the Center for Biological imaging (CBI), Institute of Biophysics, and Pei Xia for sample screening and data collection in the cryo-EM platform of PUK. We also thank Y. Chen, Z. Yang and B. Zhou for SPR technical support and X. Yu for AUC technical guidance. Work was supported by the Strategic Priority Research Program (XDB29010000), Beijing Natural Science Foundation-Haidian Primitive Innovation Joint fund (19L2008), the Key Programs of the Chinese Academy (KJZD-SW-L05), the National Key Research and Development Program (2018YFA0900801 and 2017YFC0840300), National Science Foundation of China (31800145, 31941011, 31900873 and 81520108019) and the grant from the NHC Key laboratory of Enteric Pathogenic Microbiology (Jiangsu Provincial Center for Disease Control and Prevention, EM201901). Xiangxi Wang was supported by Ten Thousand Talent Program and the NSFS Innovative Research Group (No. 81921005). Ling Zhu was supported by the Youth Innovation Promotion Association at the Chinese Academy of Sciences (2019098).

## Author contributions

K.W., L. Zhu, and Y.S. performed the experiments; K.W., L. Zhu, and X.W. solved the structure; M.L. and W.Z. developed the algorithm for receptor prediction; X.Z., L. Zhang, L.C., and G.G. provided reagents, X.W., F.Z., Z.R., and K.W. designed the study, all authors analyzed data, K.W. and X.W. wrote the paper.

## Competing interests

The authors declare no competing interests.
