## [Peer Review File · Nature Communications]

REVIEWER COMMENTS:

Reviewer #1 (Remarks to the Author):

This is a very well written, elegant and an important study on the prevalent enterovirus Enterovirus 30, causing difficult outbreaks and infections worldwide. There is still very limited information on the virus host cell interaction concerning E30 although, recently, Fc receptor was shown in two distinguished articles (groups of Coyne and Gao) to act as an essential receptor for E30. In this paper, E30 structure was studied in detail and revealed structural changes upon receptor binding. Importantly, the authors show that Fc receptor binding causes a collapse of the hydrophobic pocket indicative of uncoating already at neutral conditions, while DAF does not induce the same changes. The observed structural comparison of full and empty viruses, and the neutralizing antibodies elicited even by the empty viruses give hope of their use for effective and safe vaccine production. The structure-based in-silico algorithm to evaluate the receptor binding area is a novel tool contributing to evaluation of receptor usage of different enteroviruses.

Major point:

If I understood correctly, the authors mixed the E30 preparation that had full (50%), empty (43%) and A-particles (7%) with the receptor preparations. Why didn't they perform the binding experiments with the purified full, 143S particles from the gradient and not with a mixture of structures? How to distinguish the Fc receptor bound collapsed virions from naturally empty virions with Fc receptor possibly bound? Did the authors find binding of the receptor to the A particle or to the empty virions?

Minor points:

The authors could describe in more detail whether the naturally empty virus was without the pocket factor? The great stability of the empty virions would suggest that the pocket factor could be still inside to confer stability.

Do the authors think that using the in-silico tool, it is possible to foresee epitopes or groups of epitopes which could help in developing better and broadly acting vaccines. Perhaps combining the most relevant receptor binding information helps to choose the most important serotypes for future multivalent vaccines?

Do the authors think that the naturally produced empty viruses are a valid way to produce vaccines? Do the authors think that E30 VLPs produced in cell cultures would show similar stability as the naturally produced empty viruses?

The developed in-silico method is based on the present and future high-resolution structures which allow evaluation of receptor binding sites. Don't the authors think that the future serotypes that are formed by recombination still create a need for making further high-resolution structures in order to be evaluated for receptor usage and combating the future outbreaks?

Methods:

For how long and at what temperature and buffer conditions were the viruses allowed to bind to the receptor preparations before preparing the grids for cryo EM? This information should be added to the material and methods.

Varpu Marjomäki

Reviewer #2 (Remarks to the Author):

In the present manuscript by Wang and collaborators illustrate via a series of high resolution cryo-EM structures the mechanism by which echovirus 30 (E30) binds its receptors. In the first part, the authors describe the cell entry intermediates of E30: the procapsid (E), the full capsid (F) and the altered particle (A). This sequence of conformations is well known for many members of picornavirus family and places E30 in a wider structural context. Following this characterization, the authors determine the structure of the F-particle bound the CD55 and to the FcRn receptors. These two structures correspond to two distinct stages in the cell entry process: the attachment of the capsid to the cell and the priming the capsid for genome uncoating by releasing the pocket factor, respectively. While some of these mechanisms were described for other echoviruses, the current study offers a suite of structures that builds a comprehensive view of the molecular details involved in each of these steps.

However, the strongest point of this article is the last part, where the authors propose a structure-based in silico algorithm capable to predict receptor usage for enteroviruses. This part alone will guarantee a wide audience in the field of structural virology and strongly recommends the manuscript for publication in Nature Communication.

Minor points: A series of mistakes, typos and ambiguities must be corrected before publication.

Line 146: This type of details should be in the Methods section.

Line 158: The three types of particle (should be particles)

Line 179: α A helices - Get rid of A

Line 183: Structural basis for distinguishing the groups within the EVs

Line 242: Expand the abbreviation SCR = short consensus repeat. It's not mentioned anywhere.

Line 338-339: 1,500xg should be 1,500 x g and 4°C should be 4 °C. 120,000xg should be 120,000 x g.

Line 341: Same as above. Be consistent in the use hours or h.

Line 348: 120-kV TEM – provide microscope name

Line 352: expand the FCGRT abbreviation= Fc Fragment Of IgG Receptor And Transporter

Line 358: This part should be written with more details. How was the sample prepared: receptor mix? Incubation time and temperature?

Line 364: close should be dose

Line 374: Starting reference? Or was it ab initio

Line 379: What Relion version?

Line 388: was corrected should be were corrected

Line 416: Particle stability thermal shift assay

Line 801: kD should be kDa. In the text should add a gap between the number and °C

Line 846: In figure S2 panel d the authors should use a colour scheme varying with depth that makes more clear the features of the maps.

Mihnea Bostina

Reviewer #3 (Remarks to the Author):

The manuscript describes cryo-EM structures of Echovirus 30 (E30) in various states and E30 in complex with its attachment or uncoating receptor. Through structural comparisons with other EVs, the authors reveal serotype-specific epitopes and specific differences between the subgroups. They also deduce the molecular basis for E30 receptor recognition. Besides, they develop a structure-based in silico method for prediction of EVs' receptor usage. Overall, the structural and computational studies are coherent, and the conclusion is interesting and may bring new way for rational predictions for enterovirus receptor usage. I therefore believe this work is worthy of publication in the prestigious Nature Communications. Still, there are a few comments and

questions the authors need to address.

1. The authors are suggested to include a brief introduction on the current status of structural studies on EVs and EV-receptors, and to make it more clear on the novelty of their current study on the EV30 and EV30-receptor.
2. The current version is lack of cryo-EM data processing details. Please add a supplementary figure to illustrate the processing procedure and modify related Methods accordingly.
3. For the E30-receptor maps, the authors are suggested to show the overall map-model fitting for one asymmetric unit, and a zoom-in view on the interaction interface regions.
4. In P.6 line 187-189, the authors describe that "E30 possesses five additional continuous and "star-shaped" protrusions surrounding the mesa, characterized as specific for EV-As, although the overall structure of E30 most closely resembles those of other EV-Bs (Fig. 2a)." It is hard to recognize the mentioned feature in this figure. Please clarify where this feature locates and how it appears on the figure.
5. In the main text, the authors describe that "we determined the cryo-EM structures of E30 F-particle in complex with FcRn and CD55". However, in Table S1, it is listed that "Empty particle in complex with FcRn" and "Empty particle in complex with CD55". Please clarify which kind of E30 particles was used to obtain these two E30-receptor maps.

Minor points:

- 1) Fig. 3C is too busy, and need to be regenerated to better illustrate the location of the interactions and involved residues.
2. Table S2, third line in this table, " β C" should be "BC". Also, for Tables S2 and S3, please describe how the list of the interactions is generated.
3. In P.12, line 375, the authors describe "E30-FcRn-complex and E30-4B10-complex". Please clarify where this 4B10 come from.
4. In P. 11, line 348, please describe the machine type of the "120 kV-TEM" used.

Manuscript Title: " **Structures of Echovirus 30 in complex with its receptors inform a rational prediction for enterovirus receptor usage**"

Response to referees' comments

We appreciate the reviewers' constructive and insightful comments, and we believe that after incorporating the reviewer's suggestions the manuscript has been strengthened.

Reviewer #1 (Remarks to the Author):

This is a very well written, elegant and an important study on the prevalent enterovirus Enterovirus 30, causing difficult outbreaks and infections worldwide. There is still very limited information on the virus host cell interaction concerning E30 although, recently, Fc receptor was shown in two distinguished articles (groups of Coyne and Gao) to act as an essential receptor for E30. In this paper, E30 structure was studied in detail and revealed structural changes upon receptor binding. Importantly, the authors show that Fc receptor binding causes a collapse of the hydrophobic pocket indicative of uncoating already at neutral conditions, while DAF does not induce the same changes. The observed structural comparison of full and empty viruses, and the neutralizing antibodies elicited even by the empty viruses give hope of their use for effective and safe vaccine production. The structure-based in-silico algorithm to evaluate the receptor binding area is a novel tool contributing to evaluation of receptor usage of different enteroviruses.

We thank the reviewer for a very comprehensive evaluation of our work on E30 and for considering it an important contribution.

Major point:

If I understood correctly, the authors mixed the E30 preparation that had full (50%), empty (43%) and A-particles (7%) with the receptor preparations. Why didn't they perform the binding experiments with the purified full, 143S particles from the gradient and not with a mixture of structures?

Sorry, we did not state some of the key points as clearly as we should have. This has led to some misunderstanding, which was largely our fault. Like most enteroviruses, two predominant types of E30 particles: full particles (containing packaged RNA) and empty

particles (without RNA) could be separated using continuous sucrose density ultracentrifugation. After further investigation of E30 full particles by sedimentation coefficient analysis, we found out that the prep of E30 full particles may contain a low ratio of A-particles, which was mentioned in the manuscript (lines 143-145). Due to the robust 3D classification in cryo-EM single particle reconstruction, we used the mixtures of E30 full and empty particles for the apo viral structure studies. As expected, three types of particles: F-particle (~50%), A-particle (~7%) and E-particle (~43%) were separated when the data were processed with no-alignment 3D classification, which were reconstructed to 2.9 Å, 2.9 Å and 3.4 Å, respectively. Except for the apo virus structure determinations, the following structural investigations for E30 in complex with its receptors were performed by using the purified 143S full particles, not the mixtures. To avoid possible misunderstandings, we have provided more details on cryo-EM sample preparations in the method and have emphasized these in the revised manuscript.

How to distinguish the Fc receptor bound collapsed virions from naturally empty virions with Fc receptor possibly bound?

Thanks for your question; it's closely related with the previous point. We used the purified 143S full particles for complex preparation. In our complex structures, neither E_particle alone nor E_particle in complex with Fc receptor were observed. Although the pocket factor was released and the pocket partially collapsed in FcRn-bound F_particle structure, densities for viral RNA genome inside the capsid was clearly observed (supplementary figure 2f), indicating that RNA release may require other factors. This also suggests that FcRn bound collapsed virions are not formed from or are the same as naturally empty virions in complex with Fc receptor. In addition, protein compositions for full (VP1-VP4) and empty (VP0, VP1, VP3) particles are different. This difference can be used to find out whether FcRn bound collapsed virions are derived from natural empty virions or not. To avoid possible misunderstandings, we have provided more structural descriptions and analysis to distinguish the two types of particles.

Did the authors find binding of the receptor to the A particle or to the empty virions?

Thanks for your question. After revisiting our cryo-EM datasets carefully, we did not observe structural evidence for the binding of receptor to the A particle. Given the fact that the

exterior structures of full and empty particles are mostly indistinguishable, we believe the empty virions have the potential to interact with the receptor, which could be confirmed in a follow-up study in the near future, however this work is beyond the scope of the present manuscript.

Minor points:

The authors could describe in more detail whether the naturally empty virus was without the pocket factor? The great stability of the empty virions would suggest that the pocket factor could be still inside to confer stability.

Thanks for pointing this out. Like most enteroviruses, E30 E-particle possesses a collapsed hydrophobic pocket in VP1 without the pocket factor (shown in supplementary figure 8). More structural descriptions on this point have been provided in the revised manuscript as follows - "Like most enteroviruses, E30 F-particle possesses the hydrophobic pocket in VP1, which harbors a pocket factor. However, E- and A-particles of E30 lack a hydrophobic pocket and so they do not have a pocket factor". Regarding the relationship between particle stability and the pocket factor, a number of studies have suggested that many natural empty particles exhibit better stabilities than full particles do, like EV71, CVA16, CVA10 and E30 (Wang et.al, NSMB, 2012; Ren et.al, Nat Commun 2013, Zhu et.al, Nat Commun 2018). Similar to E30, none of these empty particles harbor the pocket factor, suggesting that the pocket factor may not confer stability on empty particles. However, many tight binders that competitively replace the pocket factor indeed stabilize mature virions, serving as antiviral inhibitors by blocking viral uncoating. These cases indicate that the pocket factor/pocket factor analogs play critical roles in stability of mature virions. Because viral particle stability is not the focus of this manuscript, discussions on the relationship between particle stability and the pocket factor are not represented in this manuscript.

Do the authors think that using the in-silico tool, it is possible to foresee epitopes or groups of epitopes which could help in developing better and broadly acting vaccines. Perhaps combining the most relevant receptor binding information helps to choose the most important serotypes for future multivalent vaccines?

Thanks for your suggestions. In-silico analysis methods have been increasingly used in different applications across many fields of biological research, such as in-silico docking, in-silico drug screening, in-silico epitope prediction, and involving many other tools as well. The method of in-silico structure-based epitope prediction (Borley et.al, PLoS One, 2013) has been extensively studied and widely used for drug/vaccine design. However, to date an in-silico tool for receptor prediction has not been developed. Our in silico algorithm for receptor prediction provides an alternative method to unveil receptor usage for human EVs quickly. These structural details on receptor binding and receptor usage revealed by in silico algorithm could rationally guide broad or multivalent vaccine design. The potential application of this in silico algorithm has been discussed in the revised manuscript.

Do the authors think that the naturally produced empty viruses are a valid way to produce vaccines? Do the authors think that E30 VLPs produced in cell cultures would show similar stability as the naturally produced empty viruses?

These are interesting questions. Structural studies suggest E30 F- and E-particles are structurally similar, with the external surface expected to be antigenically indistinguishable. Immunogenic investigations indicate E30 E-particles can elicit comparable neutralizing antibody (NAb) titers when compared to those elicited by F-particles. These two results support the candidature of natural F-particle or E-particle for vaccine development. In general, the VLP produced in cell cultures structurally resembles natural E-particle with similar particle stability. So E30 VLP may also be a viable option for vaccine development. More experimental data are needed to verify this theoretical speculation. However this work is beyond the scope of the present manuscript.

The developed in-silico method is based on the present and future high-resolution structures which allow evaluation of receptor binding sites. Don't the authors think that the future serotypes that are formed by recombination still create a need for making further high-resolution structures in order to be evaluated for receptor usage and combating the future outbreaks?

Yes, we completely agree with the reviewer. High-resolution structural information is a prerequisite for the analysis of in-silico receptor usage. As mentioned in the main text, the recent 'resolution revolution' in cryo-EM has greatly accelerated the process for determining

high-resolution structures of biological molecules. This high-resolution structural information will in turn improve the accuracy and the robustness of tools based on in-silico algorithms. These multidisciplinary methods, we believe, will usher a new direction for combating the future outbreaks.

Methods:

For how long and at what temperature and buffer conditions were the viruses allowed to bind to the receptor preparations before preparing the grids for cryo EM? This information should be added to the material and methods.

Thanks for pointing these out. More details on complex preparations have been provided in the Method section of the revised manuscript.

Varpu Marjomäki

Reviewer #2 (Remarks to the Author):

In the present manuscript by Wang and collaborators illustrate via a series of high resolution cryo-EM structures the mechanism by which echovirus 30 (E30) binds its receptors. In the first part, the authors describe the cell entry intermediates of E30: the procapsid (E), the full capsid (F) and the altered particle (A). This sequence of conformations is well known for many members of picornavirus family and places E30 in a wider structural context. Following this characterization, the authors determine the structure of the F-particle bound to the CD55 and to the FcRn receptors. These two structures correspond to two distinct stages in the cell entry process: the attachment of the capsid to the cell and the priming the capsid for genome uncoating by releasing the pocket factor, respectively. While some of these mechanisms were described for other echoviruses, the current study offers a suite of structures that builds a comprehensive view of the molecular details involved in each of these steps. However, the strongest point of this article is the last part, where the authors propose a structure-based in silico algorithm capable to predict receptor usage for enteroviruses. This part alone will guarantee a wide audience in the field of structural virology and strongly recommends the manuscript for publication in Nature Communication.

We thank the reviewer for a very comprehensive evaluation of our work on E30 and for considering it an important contribution.

Minor points: A series of mistakes, typos and ambiguities must be corrected before publication.

Line 146: This type of details should be in the Methods section.

Thank, done.

Line 158: The three types of particle (should be particles)

Thanks, corrected.

Line 179: α A helices - Get rid of A

Thanks, corrected.

Line 183: Structural basis for distinguishing the groups within the EVs

Thanks, done.

Line 242: Expand the abbreviation SCR = short consensus repeat. It's not mentioned anywhere.

Thanks, done.

Line 338-339: 1,500xg should be 1,500 x g and 4°C should be 4 °C. 120,000xg should be 120,000 x g.

Thanks, corrected.

Line 341: Same as above. Be consistent in the use hours or h.

Thanks, corrected.

Line 348: 120-kV TEM – provide microscope name

Thanks, we have added the name – FEI Tecnai Spirit.

Line 352: expand the FCGRT abbreviation= Fc Fragment Of IgG Receptor And Transporter

Thanks, we have added the full name.

Line 358: This part should be written with more details. How was the sample prepared: receptor mix? Incubation time and temperature?

Thanks for the suggestions. We have added more details in the method section of the revised manuscript as follows - “For cryo-grids preparation, the purified E30 F-particles were mixed with the E30 E-particles to obtain the E30 mixtures. The purified E30 F-particles were mixed with FcRn or CD55 (at a ratio of ~1:200) on ice for 10 seconds to obtain the E30 F-particle-receptor mixtures. Afterwards, 3 µl aliquots of the E30 mixtures or the mixture of E30 F-particle with FcRn or the mixture of E30 F-particle with CD55 were deposited onto freshly glow-discharged 400-mesh holey carbon-coated gold grid”.

Line 364: close should be dose

Thanks, corrected.

Line 374: Starting reference? Or was it ab initio

Thanks for pointing this out. The initial reference model was generated by Relion; more details have been provided in the method section of the revised manuscript.

Line 379: What Relion version?

Thanks, we have now provided information on the Relion version in the revised manuscript.

Line 388: was corrected should be were corrected

Thanks, done.

Line 416: Particle stability thermal shift assay

Thanks, done.

Line 801: kD should be kDa. In the text should add a gap between the number and °C

Thanks, corrected.

Line 846: In figure S2 panel d the authors should use a colour scheme varying with depth that makes more clear the features of the maps.

Thanks for your suggestions, done!

Mihnea Bostina

Reviewer #3 (Remarks to the Author):

The manuscript describes cryo-EM structures of Echovirus 30 (E30) in various states and E30 in complex with its attachment or uncoating receptor. Through structural comparisons with other EVs, the authors reveal serotype-specific epitopes and specific differences between the subgroups. They also deduce the molecular basis for E30 receptor recognition. Besides, they develop a structure-based in silico method for prediction of EVs' receptor usage. Overall, the structural and computational studies are coherent, and the conclusion is interesting and may bring new way for rational predictions for enterovirus receptor usage. I therefore believe this work is worthy of publication in the prestigious Nature Communications. Still, there are a few comments and questions the authors need to address.

We thank the reviewer for the perspective and constructive comments, which will help improve this manuscript.

1. The authors are suggested to include a brief introduction on the current status of structural

studies on EVs and EV-receptors, and to make it more clear on the novelty of their current study on the EV30 and EV30-receptor.

Thanks for your suggestion. Additional background information regarding the current status of the structural studies of EVs and EV-receptors have been provided in the INTRODUCTION of the revised manuscript.

2. The current version is lack of cryo-EM data processing details. Please add a supplementary figure to illustrate the processing procedure and modify related Methods accordingly.

Thanks for your suggestion. Workflow and details for cryo-EM data processing have been provided in the supplementary figure S2 and Methods section of the revised manuscript.

3. For the E30-receptor maps, the authors are suggested to show the overall map-model fitting for one asymmetric unit, and a zoom-in view on the interaction interface regions.

Thanks for your suggestion. A new supplementary figure (figure S4) has been added to show the overall map-model fitting for one asymmetric unit, and a zoom-in view of the interaction interface regions is shown.

4. In P.6 line 187-189, the authors describe that "E30 possesses five additional continuous and "star-shaped" protrusions surrounding the mesa, characterized as specific for EV-As, although the overall structure of E30 most closely resembles those of other EV-Bs (Fig. 2a)." It is hard to recognize the mentioned feature in this figure. Please clarify where this feature locates and how it appears on the figure.

Thanks for pointing this out. We have marked the "star-shaped" protrusion in the Fig. 2a and have described the protrusions in the main text and figure legends as follows - "These star-like protrusions arise from the loops near the five-fold axes, in particular the VP1 BC loop, which assumes a slightly longer and raised conformation when compared to their counterparts from other EV-Bs".

5. In the main text, the authors describe that "we determined the cryo-EM structures of E30 F-particle in complex with FcRn and CD55". However, in Table S1, it is listed that "Empty

particle in complex with FcRn” and “Empty particle in complex with CD55”. Please clarify which kind of E30 particles was used to obtain these two E30-receptor maps.

Many thanks for pointing this out. These are typos and have been corrected now. As mentioned in the method and main text, we used the purified E30 F-particles for complex preparations (shown in cryo-EM images in supplementary figure 3a). 2D classification and 3D reconstruction results show clear densities for the viral RNA genome (supplementary figure 3e), further verifying that they are F-particles.

Minor points:

1) Fig. 3C is too busy, and need to be regenerated to better illustrate the location of the interactions and involved residues.

Thanks, done.

2. Table S2, third line in this table, “ β C” should be “BC”. Also, for Tables S2 and S3, please describe how the list of the interactions is generated.

Many thanks for pointing these out. We have corrected it and more details about how the interactions were analyzed are provided in the figure legends.

3. In P.12, line 375, the authors describe “E30-FcRn-complex and E30-4B10-complex”. Please clarify where this 4B10 come from.

Many thanks for pointing this out. It should be CD55 rather than 4B10. We have submitted two closely related manuscripts on E30 to Nat Commun, one is this manuscript focusing on viral entry and the other involves neutralization mechanism of the antibody 4B10. We have corrected it in the revised manuscript.

4. In P. 11, line 348, please describe the machine type of the “120 kV-TEM” used.

We thank the reviewer’s suggestion and have added the name – FEI Tecnai Spirit here.

REVIEWERS' COMMENTS:

Reviewer #1 (Remarks to the Author):

This is a very well written, elegant and an important study on the prevalent enterovirus Enterovirus 30, causing difficult outbreaks and infections worldwide. The authors show that Fc receptor binding causes a collapse of the hydrophobic pocket indicative of uncoating already at neutral conditions, while DAF does not induce the same changes. The observed structural comparison of full and empty viruses, and the neutralizing antibodies elicited even by the empty viruses give hope of their use for effective and safe vaccine production. The structure-based in-silico algorithm to evaluate the receptor binding area is a novel tool contributing to evaluation of receptor usage of different enteroviruses.ew.

In the revised manuscript, the authors have satisfactorily answered my questions and comments. The paper is now acceptable for publication from my point of view.